# Co-Design with Rural Adolescents to Inform a School-Based Physical Activity and Social Media Literacy Intervention: A Qualitative Study

**DOI:** 10.3390/ijerph22101501

**Published:** 2025-09-30

**Authors:** Janette M. Watkins, Janelle M. Goss, Autumn P. Schigur, Megan M. Kwaiser, McKenna G. Major, Cassandra Coble, Krista Wisner, David Koceja, Vanessa M. Martinez Kercher, Kyle A. Kercher

**Affiliations:** 1Department of Kinesiology, College of Health and Human Development, Pennsylvania State University, University Park, PA 16802, USA; 2School of Public Health-Bloomington, Indiana University, Bloomington, IN 47405, USA; jangoss@iu.edu (J.M.G.); aschigur@iu.edu (A.P.S.); mkwaiser@iu.edu (M.M.K.); mckmajor@iu.edu (M.G.M.); coblec@iu.edu (C.C.); koceja@iu.edu (D.K.); vkercher@iu.edu (V.M.M.K.); kylkerch@iu.edu (K.A.K.); 3Department of Psychological & Brain Science, College of Arts and Sciences, Indiana University, Bloomington, IN 47405, USA; kwisner@iu.edu

**Keywords:** youth health, community research, intervention design

## Abstract

Cardiovascular disease remains the leading cause of death in the United States, with rural communities experiencing elevated risk. Youth in rural settings are particularly vulnerable, reporting worse health outcomes than their urban peers. The growing influence of social media has added complexity to adolescent health behaviors, particularly among youth experiencing challenges with physical and mental well-being. This qualitative study presents findings from a co-design initiative conducted with rural middle school students to examine adolescents’ views on body image, social media use, and engagement in physical activity, and to inform the development of the Hoosier Sport Re-Social intervention. Fourteen middle school students (grades 7–8) from a rural community participated in a structured co-design process spanning five sessions over nine weeks. A deductive thematic analysis was employed using Basic Psychological Needs Theory (BPNT) and Social Comparison Theory (SOCO) as guiding frameworks.

## 1. Introduction

Cardiovascular disease (CVD) remains the leading cause of death in the United States, with rural communities experiencing elevated risk due to systemic and socioeconomic challenges [1,2]. Engaging in regular physical activity (PA) is a well-documented protective factor against CVD, and establishing healthy activity patterns early in life can support long-term health trajectories. Despite these benefits, recent data indicate that only 20–28% of U.S. children and youth aged 6–17 meet the recommendation of 60 min of daily moderate-to-vigorous physical activity, with compliance dropping from about 30–40% in younger children (6–11 years) to roughly 15% in adolescents (12–17 years) [3]. Youth in rural settings are particularly vulnerable, as they report worse health outcomes than their urban peers [4,5,6,7,8]. This disparity is partly attributed to limited access to healthcare, financial constraints, and geographic isolation [9,10,11]. In rural areas, access to PA opportunities may be constrained by a lack of affordable and conveniently located sport facilities [8,12], fewer active role models [4], and reduced parental encouragement to be physically active [5].

The growing influence of social media has added complexity to adolescent health behaviors, particularly among youth experiencing challenges with physical and mental well-being [13]. While these platforms can serve as avenues for connection, self-expression, and accessing health information, they also introduce notable risks [14]. Adolescent girls appear especially vulnerable to the adverse effects of social media, where exposure to idealized portrayals and comparison-driven content is widespread [13,15]. Studies consistently show that increased engagement with social media is associated with greater body dissatisfaction, diminished self-esteem, and elevated anxiety levels, particularly among adolescent girls and young women [16,17]. These associations are observed in international samples as well [18,19]. These psychological outcomes are often intensified by platform features—such as algorithmic feeds, public reactions (e.g., likes, comments), and appearance-based content—that promote social comparison and reinforce external validation. Despite these findings, more research is needed to explore how these digital features interact with individual characteristics, including developmental stage, gender identity, and pre-existing mental health issues, in shaping adolescent well-being.

Adolescent anxiety is a growing public health concern, with emerging evidence highlighting the role of social media in its development and maintenance [20]. High levels of social media engagement have been linked to increased stress, loneliness, and feelings of inadequacy, along with sleep disturbances that can worsen anxiety symptoms [21,22,23]; cross-national work similarly implicates FOMO and related mechanisms [18,24]. Online environments often amplify social pressures, including fear of missing out (FOMO) [18,24], negative peer experiences, and the relentless pursuit of social approval—all of which contribute to elevated anxiety [21]. In contrast, regular PA offers a promising strategy for anxiety reduction [25,26]. Through the release of mood-enhancing neurochemicals like endorphins and the reduction in stress hormones such as cortisol, PA supports both physiological and emotional regulation [27,28]. Moreover, involvement in sports and recreational activities promotes a sense of achievement and nurtures social relationships, which act as buffers against anxiety [29]; these psychosocial benefits are also reflected in international studies [30,31]. For adolescents in rural areas, structured PA opportunities—such as school sports or physical education—can serve as vital outlets for stress relief and self-esteem building [30].

To tackle the intertwined challenges of physical inactivity, social media exposure, and mental health concerns in rural youth, Hoosier Sport Re-Social was created as a comprehensive, school-based intervention tailored for middle school students in rural communities. Grounded in the six-step framework of Intervention Mapping [32], the program was shaped through a participatory co-design process involving students, educators, parents, and local stakeholders. This collaborative approach was central to ensuring cultural relevance and responsiveness to the specific needs of the target population. By integrating community voices, particularly those of adolescents, the program fostered a greater sense of ownership and engagement, thereby enhancing the potential for long-term impact [33,34]. Co-design strategies emphasize inclusive decision-making, ensuring that diverse perspectives—especially those often overlooked in traditional adult-led planning—play a vital role in shaping the intervention. To structure its development and evaluation, the program also employed the Obesity-Related Behavioral Intervention Trials (ORBIT) model [35], which supports the methodical creation of behavior change interventions through iterative, evidence-informed stages.

This qualitative study presents findings from a co-design initiative conducted with rural middle school students. Through a series of participatory workshops, we examined adolescents’ views on body image, social media use, and engagement in physical activity. The study also uncovered key challenges related to implementing and scaling the intervention, offering practical insights for addressing health disparities in underserved rural settings. Findings highlight both the priorities identified by youth participants and emergent themes related to the impact of social media, barriers to physical activity, and mental health concerns. These insights contribute to the development of inclusive, community-informed approaches aimed at supporting adolescent health and well-being. By integrating youth perspectives into the early design stages, this work uniquely bridges digital wellness and physical activity within a rural context, a combination rarely addressed in prior interventions. The insights generated not only inform refinement of Hoosier Sport Re-Social but also contribute broadly to innovative, community-informed strategies for adolescent health promotion. We hypothesized that rural adolescents would identify both social media use and limited access to physical activity opportunities as key influences on their well-being, and that co-design methods would reveal feasible, context-specific strategies to address these challenges.

## 2. Materials and Methods

### 2.1. Conceptual Framework

The development of the program was grounded in several complementary theoretical frameworks, including Self-Determination Theory (SDT) [36] and its mini-theories—Basic Psychological Needs Theory (BPNT) and Goal Contents Theory (GCT)—as well as Social Cognitive Theory (SCT) [37] and Social Comparison Theory (SOCO) [38]. BPNT provided a foundational lens for both intervention design and qualitative analysis by emphasizing the fulfillment of three core psychological needs: autonomy, competence, and relatedness. These elements are critical to fostering intrinsic motivation and supporting long-term behavior change. The intervention was intentionally structured to give participants a sense of choice and agency (autonomy), opportunities to build skills (competence), and experiences that nurtured peer and mentor relationships (relatedness). GCT further enriched this approach by highlighting the importance of intrinsic goals—such as personal well-being and self-improvement—over extrinsic motivators like appearance or social validation. This theoretical alignment supported the program’s emphasis on body positivity and cultivating genuine enjoyment of physical activity.

Figure 1 depicts the hypothesized pathway from guiding levels of influence (individual, interpersonal, community; SCT) to intervention strategies (sport participation, skill development with goal setting/monitoring, structured social support, education in physical and social-media literacy, and take-home activities) and then to theoretical mediating processes (satisfaction of competence, autonomy, and relatedness; BPNT within SDT). SCT constructs (self-efficacy, role modeling, social support) inform delivery at each level; GCT orients content toward intrinsic goals (well-being, mastery, enjoyment) rather than extrinsic (appearance/validation); and SOCO is addressed via media-literacy and norms-reflection components designed to reduce harmful upward comparison. Solid arrows indicate proposed causal links (levels → strategies → needs → behavior), and dotted arrows indicate mediating processes (need satisfaction) and cross-cutting influences (social comparison). These same constructs defined the deductive codebook and framework analysis matrix used in qualitative analysis.

### 2.2. Participants and Procedures

Adolescents were recruited from WRV Middle School using predetermined inclusion criteria. Eligible participants were 7th and 8th grade students who completed digital assent and obtained parental consent, both administered via Qualtrics to facilitate secure and efficient data collection. Exclusion criteria included students with poor school attendance or academic performance, as the school requested that students who needed classroom time not be pulled from instruction. All study procedures received approval from the Indiana University Institutional Review Board (#21473). A total of 14 students participated in the co-design workshops. The sample included seven girls: White (n = 4, 57.1%), Hispanic (n = 2, 28.6%), and Black (n = 1, 14.3%). The seven boys included: White (n = 5, 71.4%) and Hispanic (n = 2, 28.6%). Students engaged in a structured co-design process aimed at collaboratively informing the intervention’s content, delivery, and evaluation to reflect local community needs [39]. The co-design process spanned five sessions over a nine-week period. The first three sessions were held weekly, followed by two consecutive sessions that allowed for iterative feedback and refinement of ideas. Each session was designed with specific goals (see Figure 2) and began with activities focused on identifying community health priorities, exploring barriers to physical activity and social media literacy, and setting the foundation for the intervention’s core objectives. Facilitators guided participants through discussions and collaborative exercises that encouraged open dialogue and active input on key program elements.

Each co-design session was held in a private classroom setting, with one to three research team members present to support the child participants. Sessions began with a short icebreaker activity (e.g., a warm-up relay or a question-and-answer game with a ball) to build rapport and foster engagement, followed by a brief recap of the previous session. The facilitator then introduced the session’s focus, leading a large-group discussion before transitioning into small-group tasks such as identifying challenges, generating ideas, or ranking priorities. To close each session, participants reconvened to share takeaways, reflect on key insights, and offer personal stories related to the discussion topics. Immediately afterward, participants were invited to complete an anonymous feedback survey accessed via a QR code, allowing them to share their perspectives on the session experience. The research team (n = 4) conducted debriefing meetings after each session to review participant input, assess emerging themes, and adapt future session plans. In addition to adolescent participants, an expert advisory committee—comprising a teacher, two parents, a school board member, and the school principal—was convened to provide adult perspectives. The committee did not contribute data for analysis but reviewed session materials, monitored feasibility, and offered feedback throughout implementation.

### 2.3. Data Analysis

Qualitative data analysis drew upon insights generated during the 9-week co-design process with rural adolescents, with sessions spaced to support continuity and allow for reflection between meetings. A deductive thematic analysis was employed, using Basic Psychological Needs Theory (BPNT) and Social Comparison Theory (SOCO) as initial guiding frameworks. Session transcripts, along with observation notes for contextual depth, served as the primary data sources. Two independent coders (J.G., J.W.) reviewed and coded the transcripts to identify recurring patterns and emergent subthemes. The research team then applied a framework analysis approach to systematically organize and interpret themes within a matrix structure, aligning them with the study’s theoretical foundations—including Self-Determination Theory (SDT), Goal Contents Theory (GCT), Social Cognitive Theory (SCT), and SOCO. This analytical strategy provided a structured lens to examine how psychological needs, intrinsic goal pursuit, social dynamics, and comparison processes influenced participants’ contributions and experiences throughout the co-design process.

To complement these qualitative sources, participant feedback surveys administered after each session were analyzed descriptively and thematically. Open-ended responses were coded alongside workshop transcripts to capture adolescents’ direct reflections on feasibility, acceptability, and perceived relevance of proposed strategies. Closed-ended survey items were summarized quantitatively (e.g., frequencies, proportions) to provide contextual indicators of satisfaction and engagement. These multiple data streams were integrated during analysis to triangulate findings and strengthen interpretation.

Thematic saturation was reached when the data fully captured the study’s core constructs—autonomy, competence, relatedness, intrinsic and extrinsic motivation, and social comparison. Subthemes were generated through an iterative coding process, starting with open coding to identify salient patterns and participant language, followed by organizing these codes into broader categories that reflected the study’s theoretical framework. To strengthen credibility and reduce potential bias, multiple researchers coded the data independently. When coding disagreements occurred, they were first documented, then discussed in structured resolution meetings, and if consensus could not be reached, a third researcher adjudicated to finalize the code. Saturation was determined when no new themes, subthemes, or meaningful variations emerged during analysis. The team evaluated saturation by noting the recurrence of concepts across participants, the consistency of findings within the theoretical lens, and the overall richness and coherence of the emerging interpretations.

To support theme refinement, affinity diagramming was used to visually group related codes and identify overarching patterns across the data. A third team member (J.M.G.) contributed to the development and validation of the final codebook (see appendix), which was subsequently reviewed by additional researchers (J.M.W., A.P.S. and M.M.K.) to ensure consistency and enhance analytic rigor. Multiple rounds of coding enabled ongoing refinement of categories and subthemes. Insights from prototype testing sessions were integrated to assess feasibility and acceptability, informing the intervention’s practical design. Discrepancies in interpretation were addressed through collaborative workshops, which facilitated consensus among team members and contributed to the finalization of a pilot-ready intervention protocol and implementation strategy grounded in participants’ lived experiences and priorities.

The research team acknowledges that our professional and personal backgrounds influenced the design, facilitation, and interpretation of the co-design process. The two primary facilitators (J.M.W. and J.M.G.) had previously worked in the rural region on multiple school- and community-based projects and were familiar with the community context, which supported rapport and trust with participants. This familiarity may also have shaped assumptions during data collection and analysis. Other team members (A.P.S. and M.M.K.) contributed expertise in youth physical activity, mental health, and qualitative methodology, which informed both coding and interpretation. To reduce potential bias, we employed independent coding, peer debriefing, and iterative consensus meetings to critically reflect on how our positionalities influenced the analytic process.

## 3. Results

### 3.1. Sample

A total of 14 middle school students (grades 7–8) from the target rural community participated in the co-design process, with a mean age of 13 years (SD = 1.3). The group was evenly divided by gender, including seven girls and seven boys. Thematic findings were interpreted through the lenses of Basic Psychological Needs Theory (BPNT)—focusing on autonomy, competence, and relatedness—and Goal Contents Theory (GCT), which distinguishes between intrinsic and extrinsic motivation. Additionally, Social Comparison Theory (SOCO) was used to contextualize participants’ reflections on peer influence and perceived social norms. Together, these frameworks provided a robust structure for understanding behavioral change processes as described by the adolescent participants.

### 3.2. Theme 1: Autonomy

The child co-designers primarily emphasized autonomy in their social media usage, highlighting a desire for greater freedom from parental controls. However, others expressed that these restrictions helped them be more physically active.

“Yeah. When they like put boundaries on my phone, I’m like, why can’t I have privacy?” -8th grade female

“Like if I didn’t have boundaries, I wouldn’t be in basketball because I would be addicted to Fortnight. Now I choose basketball over Fortnight.” -8th grade male

### 3.3. Theme 2: Relatedness

The child co-designers emphasized relatedness primarily through their PA experiences, particularly in their interactions with teammates. These experiences were both positive and negative.

“Like knowing that your teammates are behind you [makes it more fun].” -7th grade female

“[…] I had a lot of drama when I was in basketball. Like a lot of people didn’t want me to play.” -8th grade female

Children also highlighted the influence of their parents and other adults on their participation in PA, emphasizing how role modeling encouraged them to stay active.

“My mom motivates me because she’s always been supporting me. And I look up to her because she’s a hardworking woman, and she raised all these kids by herself and all that.” -8th grade male

“[My grandfather] has given me some good wording. He helps me with like pitching. And one saying that always like if I’m on the pitcher mound and I’m like stressing or something, he says one thing that always gets me to go to back to do it.” -8th grade male

### 3.4. Theme 3: Competence

Children discussed feelings of competence and how these perceptions influenced their participation in PA. Themes of competence often emerged when they shared experiences of receiving negative feedback and how they perceived their own skill level and physical ability.

“I mean, of course I want to be better. But like, I mean, I just want to know what I did wrong [to work on it].” -7th grade female on receiving negative feedback

“Like if other people don’t think that I’m good, I think my, like I can think myself as good, so I won’t quit.” -8th grade female

### 3.5. Theme 4: Extrinsic Motivation

The fourth theme emphasized by the co-designers was extrinsic motivation, particularly the impact of parental influence on their performance. Children often expressed feeling external pressure when their parents were present at sporting events.

“Yeah, I play better when my whole family is not there. It’s weird. Like literally, I’m into tears when I lose a game because my family is like telling me I have to do better and I’m not good enough.” -8th grade female

“No, because my parents like criticize me after games. They just like tell me what I did wrong and don’t tell me what I did good.” -8th grade male

When discussing their primary motivations for using social media, females reported being influenced by negative extrinsic factors, leading them to spend more time on platforms like TikTok and Snapchat. In particular, negative peer interactions made them feel compelled to defend themselves in front of their peers.

“So, then I post a TikTok about her that wasn’t, okay, it was directed towards her, but I kept telling her it wasn’t. And then, so she posted one about me. And then I’m petty, and then I do it back.” -8th grade female

“And then I text her on Snap, and then I was like, well, why are you like talking to me? Like I don’t really care. And then a girl named [name] …… told me that she’s going to beat me up. […] and I don’t even know who she is.” -8th grade female

### 3.6. Theme 5: Intrinsic Motivation

In contrast to extrinsic motivators, the child co-designers also highlighted intrinsic motivation for participating in physical activity, particularly as a way to regulate emotions and find relief from stress. These motivations reflect intrinsic regulation because the activity is valued for its internal rewards—such as enjoyment, emotional release, and a sense of personal well-being—rather than for external outcomes or approval from others.

“Because like you’re just so excited to get out there and like…just play […] the stress of [school] doesn’t matter.” -8th grade male

“I do [basketball] to distract myself from everything going on at home.” -8th grade female

Children also highlighted intrinsic reasons for using social media, such as avoiding responsibilities, coping with negative emotions, and simply relaxing.

“[…] like it gets me distracted […] and it’s kind of better to [use social media] than talk to my family and actually expressing my feelings to them.” -8th grade female

“Oh, because there’s everything on it, and I can just do everything on my, on a phone.” -7th grade female on why she chooses Snapchat to relax

### 3.7. Theme 6: Social Comparison

Social comparison was the most frequently discussed theme among the co-designers. Children often highlighted comparing themselves to their peers and feeling judged, particularly in relation to both social media usage and PA participation.

“They think they’re not good enough [to play soccer]. Some think they might be too fat or something compared to the [other players seen on social media]”-8th grade female

“[Girls on Snapchat] have to get up like two hours early just to do their hair and makeup.” -8th grade male

Females also emphasized concerns about their physical appearance and feeling judged, particularly by their male peers.

“So, my brother was like, you’re not going to get a boyfriend. You’re too fat. And I was like, oh. So, I get a boyfriend, and he breaks up with me. And I’m like … why did he break up with me? He goes, because you’re fat. So, yeah.” -7th grade female

“When, like when you’re in like a game and there’s like a bunch of boys, they could like shout something out [about your body] while you’re playing on court […] or post something about you [on social media]. That’s what happened to me.” -8th grade female.

While these concerns were more frequently and intensely described by female participants, some boys also reported pressures related to appearance and social standing, though often framed in terms of athletic performance, body size, or comparisons with peers on social media.

### 3.8. Intervention Design Implications

The intervention was designed to strengthen core elements of Basic Psychological Needs Theory (BPNT) and Goal Contents Theory (GCT)—specifically autonomy, competence, relatedness, and intrinsic motivation—while also aiming to mitigate the effects of negative social comparison. The program’s two primary goals are to: (1) increase adolescents’ daily physical activity and (2) decrease their daily social media use. Hoosier Sport Re-Social will be implemented in rural middle schools through a dual-delivery model: health classes will focus on body positivity and social media literacy, while physical education sessions will emphasize physical literacy and skill-building for active living. To support psychological needs, the curriculum will promote student autonomy through choice-based activities, foster relatedness through collaborative group work, and build competence via progressive skill development. Efforts to reduce harmful social comparisons will be embedded in lessons that cultivate critical media literacy and encourage body appreciation.

To illustrate the interconnections among the six themes and the study’s guiding theoretical frameworks, Figure 3 presents a conceptual model. The figure highlights how factors such as self-efficacy, role modeling, and social support influenced adolescents’ outcome expectations, goals, and behaviors. Within this structure, Basic Psychological Needs Theory (BPN) provides the foundation for understanding autonomy, competence, and relatedness, while Goal Contents Theory (GCT) distinguishes between intrinsic and extrinsic motivations. Social Cognitive Theory (SCT) contributes constructs such as self-efficacy and social support, and Social Comparison Theory (SOCO) captures the influence of upward comparison as both a challenge and a mediator. Together, this integrated framework illustrates how psychological needs, motivational processes, and comparison dynamics shaped participants’ experiences and informed the design of Hoosier Sport Re-Social.

## 4. Discussion

Authors The present study qualitatively explored adolescents’ views on physical activity, social media use, and body image through a co-design process that informed the development of the Hoosier Sport Re-Social intervention. As hypothesized, adolescents identified both social media use and limited access to PA opportunities as key influences on their well-being, and the co-design process revealed feasible, context-specific strategies to address these challenges. The findings highlight the intricate relationship between psychological needs (autonomy, competence, relatedness), motivational factors (intrinsic vs. extrinsic), and social comparison in influencing youth engagement in PA and overall mental health. This study reinforces the value of participatory co-design as an evidence-informed method for creating youth-centered interventions that address both physical activity behaviors and psychosocial outcomes [34,39]. Five key themes emerged: (1) Adolescents desired more autonomy in navigating social media, though many recognized that parental limits could help support PA engagement; (2) peer relationships played a critical role, with both social support and exclusion influencing participation; (3) perceptions of competence and external feedback impacted motivation, with some youth demonstrating resilience despite negative input; (4) while extrinsic motivators like parental expectations in sport contributed to stress, intrinsic motivations positioned PA as a coping tool; and (5) social comparison—particularly among girls—intensified concerns about body image. Collectively, these findings underscore the importance of addressing both social and psychological factors when designing adolescent health interventions.

### 4.1. Psychological Needs and Physical Activity

Key findings emphasize that the basic psychological needs of autonomy, competence, and relatedness are central to shaping children’s PA experiences and participation. This connection is well-supported in prior research [40,41,42]. For instance, longitudinal studies indicate that higher autonomy in PA is associated with stronger motivation [43], and autonomy support in physical education (PE) settings can foster intrinsic motivation for leisure-time activity [44]. Competence is also critical, as motor skill proficiency enhances perceived competence [45], which in turn predicts PA engagement from middle to high school [46]. Relatedness similarly impacts PA involvement; supportive relationships with peers and teachers in PE environments can enhance motivation, enjoyment, and psychological well-being [47]. Children in the current study echoed these findings, describing how both supportive and exclusionary interactions with peers and adults influenced their willingness to participate in PA. Collectively, these results—and the broader literature—underscore the need to design PA interventions that support children’s psychological needs.

### 4.2. Psychological Needs and Social Media Use

While psychological needs are well-studied in the context of physical activity, their role in social media use among youth remains underexplored. Emerging evidence suggests that although excessive screen time is associated with negative emotional outcomes, these effects may be mitigated when social media use meets psychological needs [48]. In this study, participants discussed using social media for self-expression, connection, and emotional relief, aligning with research indicating that psychological needs influence adolescents’ online behaviors [49]. However, recent findings also suggest that social media can frustrate these same needs, depending on the platform features and social context [50]. Notably, youth in this study expressed a desire for greater autonomy in their social media use, yet some acknowledged difficulty in regulating their usage. These perspectives reflect broader research showing that excessive parental control can undermine autonomy and increase the risk of problematic technology use [51]. More research is needed to explore how to balance parental guidance with adolescent autonomy to support digital well-being.

### 4.3. Intrinsic and Extrinsic Motivation in Physical Activity

The study also highlighted the significance of both intrinsic and extrinsic motivation in shaping children’s PA experiences. Prior research has shown that intrinsic motivation is positively associated with greater PA participation among adolescents, both during the week and on weekends [52,53]. A unique aspect of the present findings is that students most frequently identified emotional regulation—such as stress relief and coping with negative emotions—as a primary intrinsic driver of PA engagement. Although past studies suggest that intrinsic motivators are common across genders, with adolescent boys somewhat more likely to cite emotional regulation as a reason for PA participation, this study found that both boys and girls described using PA as an outlet for managing emotions [54,55]. This finding suggests a need for more nuanced investigation into how gender may influence the emotional motivations behind youth physical activity.

Extrinsic motivators also emerged prominently, particularly regarding the influence of parents. Participants reported experiencing both support and pressure from parents, aligning with previous work showing that the motivational climate created by caregivers can significantly affect a child’s engagement in PA. For example, children aged 9–14 whose parents emphasized enjoyment, effort, and self-improvement demonstrated greater intrinsic motivation, whereas those whose parents focused on winning, performance comparisons, or avoiding mistakes were less likely to be intrinsically motivated [56]. These distinctions matter, as early motivational experiences—particularly those shaped by parenting styles—may influence children’s long-term relationship with physical activity. A recent systematic review reinforces this point, noting that parents who foster autonomy, emphasize task-oriented goals, and maintain supportive parent–child relationships are more likely to encourage sustained PA engagement in their children [57]. Overall, these findings emphasize the critical roles of both emotional drivers and parental influence in supporting youth motivation for PA. Future research should continue to explore how intrinsic and extrinsic motivational factors evolve over time and contribute to lifelong PA habits.

### 4.4. Motivations for Social Media Use

During the co-design sessions, children described a range of intrinsic and extrinsic motivations for engaging with social media. Extrinsic drivers included peer-related challenges, which often led them to platforms such as TikTok and Snapchat to maintain or navigate social connections. Intrinsic motivations, on the other hand, centered on using social media as a means of emotional escape or relaxation. While research on extrinsic motivators in middle school populations remains limited, studies with college students have shown that intrinsic factors—such as entertainment, passing time, and stress relief—are strong predictors of problematic social media use. Among extrinsic motivators, only social engagement has been linked to social media addiction [58]. Consistent with these findings, participants in the current study most often cited connecting with friends or managing difficult emotions as reasons for their social media use. Further, existing research with adolescents aged 10 to 17 has found that low emotional regulation, increased procrastination, and high stress levels are associated with problematic use patterns, such as frequent or passive scrolling [19]. These findings point to the need for future research to examine how various motivational factors interact and potentially contribute to unhealthy or excessive social media behaviors in younger populations.

### 4.5. Social Comparison and Body Image

The final key finding of this study underscores the influence of social comparison—particularly among female participants—on body image concerns. Prior research has shown that social media use can negatively affect adolescents’ body image, with visually driven platforms like Instagram amplifying unrealistic beauty standards through appearance-based comparisons [18,59,60]. In this study, girls shared feelings of self-consciousness related to both their appearance and physical abilities, shaped by peer dynamics online and in-person. Notably, several participants expressed discomfort about male peers posting or commenting on their bodies via social media and reported that perceived judgment from male classmates impacted their willingness to engage in physical activity. These findings are consistent with previous studies involving adolescents aged 15 to 16, where fear of body-related judgment was a common barrier to sport participation among girls [61]. Such concerns may be intensified in activities stereotypically viewed as masculine [62]. However, other research with 16- to 19-year-olds indicates that involvement in organized sports is associated with greater life satisfaction, largely due to enhanced body appreciation [31]. These insights highlight the importance of designing body image–focused interventions that not only promote physical activity among girls but also address the social contexts—both digital and interpersonal—that shape their participation.

Beyond these findings, gender differences warrant further consideration. Biologically, pubertal transitions can heighten body awareness and sensitivity to evaluation, with earlier timing in girls linked to adverse psychological outcomes and greater body image concerns [63]. Hormonal changes during adolescence may also interact with mood regulation, contributing to higher rates of anxiety among girls compared to boys [64]. Socioculturally, adolescent girls often face stronger appearance-focused norms—online and in sport—where visual, comparison-based platforms and media exposure are associated with body dissatisfaction [65,66,67]. Gendered expectations in sport can also discourage girls from participating in activities perceived as masculine or competitive, adding barriers to engagement [68]. Psychologically, girls are more likely to internalize appearance-related feedback, which can undermine perceived competence and autonomy, reinforcing cycles of low motivation and avoidance. These experiences also intersect with peer dynamics, where negative comments or judgments about appearance may amplify vulnerability to social comparison. Taken together, these biological, sociocultural, and psychological factors help explain why girls in our study described heightened concerns about body image, judgment from peers, and social media pressures compared to boys. Addressing these intersecting influences is critical, as they shape not only immediate participation in PA but also long-term trajectories of health and self-perception. Our findings underscore the importance of designing interventions that incorporate body positivity, critical media literacy, and supportive peer and adult relationships to reduce harmful comparison and promote confidence, particularly among girls.

### 4.6. Implications for Intervention Development

The key findings of this study—highlighting the importance of supporting basic psychological needs, fostering intrinsic motivation, and minimizing harmful social comparison—will directly inform the development of a school-based intervention focused on adolescent well-being. This initiative builds upon earlier work with the original Hoosier Sport program, a PE-based intervention that demonstrated strong feasibility in a rural middle school context and showed promising effects on daily physical activity and need satisfaction [69,70]. Expanding on this foundation, Hoosier Sport Re-Social will engage students through both physical education and health classes, adopting a holistic strategy that integrates physical activity promotion, media literacy education, and body image enhancement. The intervention aims to equip students with critical thinking skills to navigate social media more mindfully, thereby reducing the negative impact of appearance-based comparison and validation-seeking behaviors. Health class lessons will emphasize body positivity and self-appreciation, while PE sessions will provide inclusive, skill-building opportunities designed to increase physical literacy and engagement. A particular emphasis will be placed on creating a supportive environment where all students—especially girls—feel confident and motivated to participate in physical activity.

### 4.7. Limitations

This study has several limitations. The use of a small, convenience-based sample (n = 14) may have introduced selection bias and limits the generalizability of findings beyond the specific rural community studied. Although the sample size is consistent with prior community-based participatory research efforts [34,71], it restricts the range of perspectives captured. Accordingly, we frame our claims in terms of transferability rather than generalizability, providing thick description of the school, community context, and workshop procedures to enable readers to assess fit to other settings. These findings may be transferable to rural middle schools with similar structural features—e.g., limited access to affordable sport facilities, transportation barriers, and comparable social-media use patterns—particularly where school-based PA and health classes are the primary venues for programming. Future studies should aim to recruit larger and more demographically diverse samples and consider alternative recruitment strategies to improve representation and inclusivity. We also acknowledge that the study was shaped by the research team’s positionality and backgrounds, which may have influenced facilitation and analysis, despite steps taken to reduce bias through independent coding and consensus-building. In addition, because the study relied primarily on adolescent voices and session transcripts, without triangulation across multiple data sources or stakeholder groups, findings should be interpreted with caution.

Despite these limitations, the co-design sessions conducted as part of Hoosier Sport Re-Social were instrumental in collaboratively developing a school-based intervention and implementation plan. By engaging adolescents as active partners in the research process, the sessions drew on their lived experiences to inform the design of behavior change strategies and program delivery. This approach aligns with established participatory research principles [72,73], promoting youth ownership and ensuring the intervention was responsive to their unique needs and priorities [74,75]. Participant input was particularly valuable in addressing critical challenges such as limited access to physical activity opportunities and concerns surrounding social media use [76], thereby enhancing the relevance, feasibility, and potential sustainability of the intervention in school settings.

## 5. Conclusions

This qualitative study provides valuable insights into the complex interplay between psychological needs, motivational factors, and social comparison in shaping rural adolescents’ engagement with physical activity and social media. Through a participatory co-design process, we identified six key themes that highlight the importance of autonomy, competence, and relatedness in both physical activity participation and social media use among middle school students. The findings underscore the critical role of intrinsic motivation in promoting healthy behaviors while also revealing the potentially harmful effects of extrinsic pressures and social comparison, particularly among female participants.

The co-design approach proved instrumental in developing a culturally relevant and responsive intervention that addresses the unique needs of rural youth. By engaging adolescents as active partners in the research process, we ensured that their voices and lived experiences directly informed the development of Hoosier Sport Re-Social. Methodologically, this work demonstrates the feasibility and added value of youth-centered co-design in rural school settings, offering a transferable model for participatory intervention development where resources are limited and local context is paramount. This collaborative methodology not only enhanced the relevance and feasibility of the intervention but also promoted youth ownership and engagement, which are essential for long-term success.

To strengthen future work, school-based interventions should (1) prioritize supports for autonomy, competence, and relatedness; (2) integrate critical media literacy and body image content—particularly responsive to girls’ experiences; (3) involve parents/caregivers to foster supportive home climates; and (4) include implementation supports for teachers/coaches (e.g., brief training and low-burden materials). Methodologically, studies should incorporate objective measures of PA and digital media behavior (e.g., wearables, passive sensing) and pre-specified implementation outcomes (acceptability, feasibility, fidelity) to inform scale-up. Additionally, the 9-week duration and limited number of sessions may constrain observation of longer-term behavioral or motivational change. Future research should test longer formats (e.g., semester- or year-long with booster sessions) and follow-up assessments (e.g., 3–6 months) to evaluate maintenance. A feasible next step is a hybrid Type 2 pilot or stepped-wedge trial across multiple rural schools to examine preliminary effectiveness (PA, body image, anxiety) alongside implementation, with gender-responsive modules and equity-focused adaptations for rural resource constraints.

The study’s findings have important implications for designing school-based interventions that address both physical activity promotion and digital wellness. Future interventions should prioritize supporting basic psychological needs, fostering intrinsic motivation, and creating supportive environments that minimize harmful social comparison. Particular attention should be paid to addressing the unique challenges faced by adolescent girls, including body image concerns and peer judgment in both digital and physical activity contexts. While this study was conducted in a specific rural community with a small sample size, the insights gained contribute to the broader understanding of adolescent health behaviors and intervention development. Future research should explore these themes in larger and more diverse populations while continuing to employ participatory approaches that center youth voices in the design and implementation of health interventions.

## Figures and Tables

**Figure 1 ijerph-22-01501-f001:**
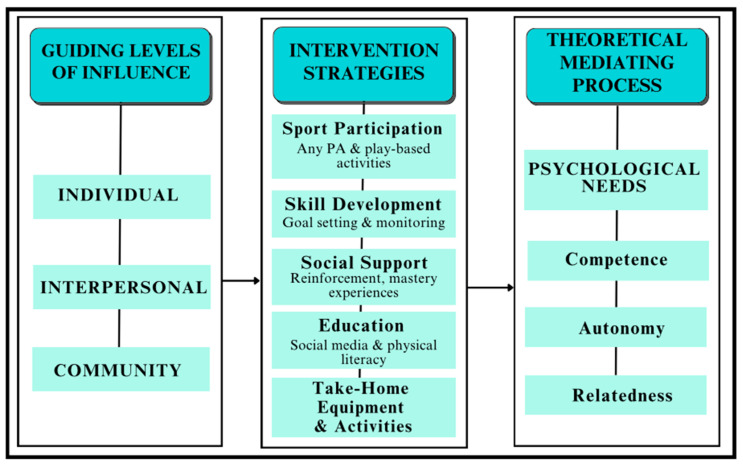
Conceptual Framework.

**Figure 2 ijerph-22-01501-f002:**
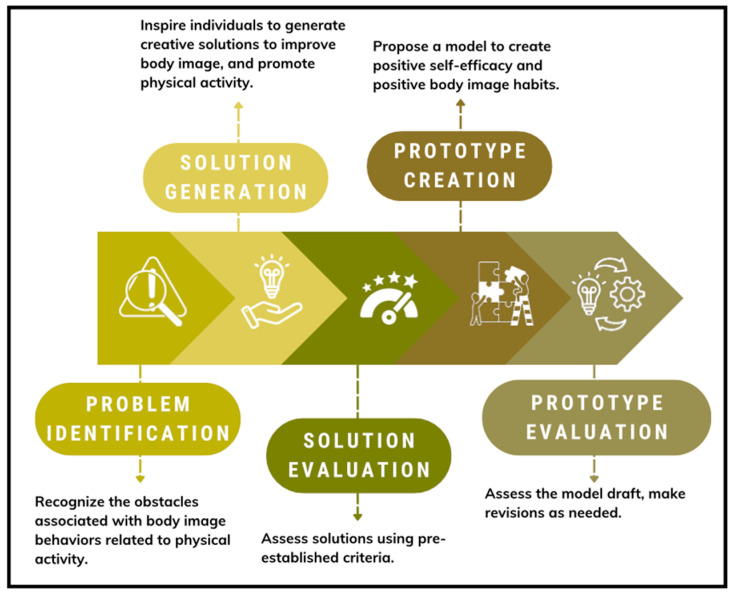
Co-Design Sessions.

**Figure 3 ijerph-22-01501-f003:**
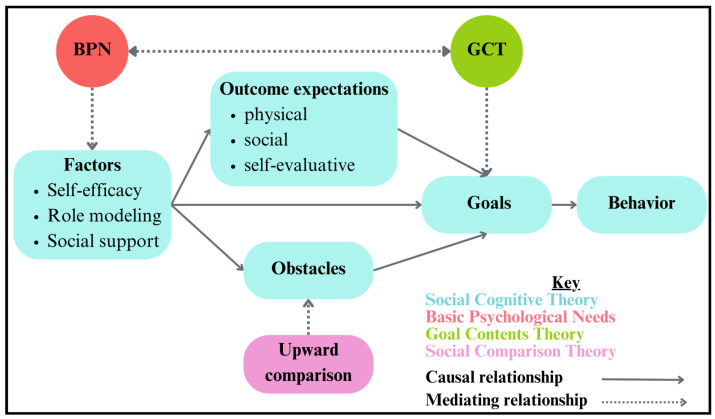
Integrated Theoretical Model of Adolescent Experiences.

## Data Availability

The raw data supporting the conclusions of this article will be made available by the authors on request.

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
