# Peer review of "Co-Design with Rural Adolescents to Inform a School-Based Physical Activity and Social Media Literacy Intervention: A Qualitative Study"

_ijerph, 2025, doi:10.3390/ijerph22101501_

Round 1
Reviewer 1 Report
Comments and Suggestions for Authors
Abstract
The abstract is too long. According to the MDPI, it should be limited to 200 words.
The six themes in lines 25-30 are not necessary in the abstract, but it should be shorter.
Introduction
The original contribution of the research is not clearly stated. While the literature review is good, it should clearly state how the study introduces innovations to the field.
There is no research question or hypothesis. The purpose is present, but the hypothesis is not stated.
Abbreviations are sufficiently explained.
The sentence in lines 38-39 states "3." The reference is from 2018; this number may have changed.
In the sentence in lines 60-62, details such as age and gender should be included when explaining the relationship between anxiety and social media.
Method
The sample size is too small (n=14). Why were 14 people selected?
A power analysis was not conducted.
More information could have been included regarding sample selection and the demographic characteristics of the participants (socioeconomic status, etc.). Exclusion criteria were not provided.
It is unclear how participant feedback was analyzed.
What is the feedback survey and what did it ask? Where was the feedback used?
Findings
There are six themes; numerical data can be added to the findings.
Tables and figures can be added.
Discussion
The discussion becomes repetitive.
The findings are well supported by the literature, but since there is no hypothesis, it is not stated whether the hypothesis was confirmed or not.
Gender is an important finding, but its discussion is very superficial; its biological, sociocultural, and psychological aspects should be explored more deeply.
The limitations section is good, but the small number of 14 participants is not emphasized.
Conclusion
Recommendations should be included.
The study's 9-week duration and limited number of sessions may limit the ability to observe long-term behavioral changes or motivational effects. These should be listed as limitations, and the feasibility of a long-term study should be included in the recommendations.
Author Response
Reviewer Comment: The abstract is too long. According to the MDPI, it should be limited to 200 words.
Response: Thank you for this suggestion. However, the current abstract is 197 words long and fits within the word limit. Please see our additional response below regarding edits to abstract length.
Reviewer Comment: The six themes in lines 25–30 are not necessary in the abstract; it should be shorter.
Response: We appreciate the reviewer’s feedback. While the six themes represent the central findings of our qualitative work, we have removed them from the abstract to address concerns regarding abstract length and to align with the reviewer’s previous comments.
Reviewer Comment: The original contribution of the research is not clearly stated. While the literature review is good, it should clearly state how the study introduces innovations to the field.
Response: Thank you for this suggestion. We have added the following to the end of the introduction (lines 110–115): “By integrating youth perspectives into the early design stages, this work uniquely bridges digital wellness and physical activity within a rural context, a combination rarely addressed in prior interventions. The insights generated not only inform refinement of Hoosier Sport Re-Social but also contribute broadly to innovative, community-informed strategies for adolescent health promotion.”
Reviewer Comment: There is no research question or hypothesis. The purpose is present, but the hypothesis is not stated.
Response: We have added a clear hypothesis to the end of the introduction (lines 115–118): “We hypothesized that rural adolescents would identify both social media use and limited access to physical activity opportunities as key influences on their well-being, and that co-design methods would reveal feasible, context-specific strategies to address these challenges.”
Reviewer Comment: Abbreviations are sufficiently explained.
Response: Thank you.
Reviewer Comment: The sentence in lines 38–39 states “3.” The reference is from 2018; this number may have changed.
Response: We thank the reviewer for pointing this out. We have updated the statement with the most recent data from the 2024 U.S. Report Card on Physical Activity for Children and Youth (Physical Activity Alliance, 2024). The revised text now states (lines 38–39): “Despite these benefits, recent data indicate that only 20–28% of U.S. children and youth aged 6–17 meet the recommendation of 60 minutes of daily moderate-to-vigorous physical activity, with compliance dropping from about 30–40% in younger children (6–11 years) to roughly 15% in adolescents (12–17 years) [3].”
Reviewer Comment: In the sentence in lines 60–62, details such as age and gender should be included when explaining the relationship between anxiety and social media.
Response: We appreciate this suggestion and have revised the sentence to specify age and gender details. The revised text now states (lines 60–62): “Studies consistently show that increased engagement with social media is associated with greater body dissatisfaction, diminished self-esteem, and elevated anxiety levels, particularly among adolescent girls and young women [17,18].”
Reviewer Comment: The sample size is too small (n = 14). Why were 14 people selected?
Response: We thank the reviewer for raising this point. As this was a qualitative co-design study, the aim was not to achieve statistical generalizability but to generate rich, contextual insights to inform intervention development. A sample of 14 participants is considered appropriate for participatory workshops, as it allows for manageable group dynamics while ensuring that diverse perspectives can be shared. This size aligns with best practices in qualitative research, where depth of discussion and quality of interaction are prioritized over large numbers (Braun & Clarke, 2013). In addition, co-design methodology often employs small, focused groups to maximize active engagement and iterative feedback (Sanders & Stappers, 2008).
Reviewer Comment: A power analysis was not conducted.
Response: We appreciate the reviewer’s observation. Because this study was qualitative and exploratory in nature, the primary aim was to generate formative insights through co-design workshops rather than to test hypotheses with inferential statistics. For this reason, a power analysis was not applicable. Instead, we followed established guidance for qualitative and co-design research, which emphasizes data richness, diversity of perspectives, and group interaction over statistical sample size determination (Braun & Clarke, 2013; Sanders & Stappers, 2008).
Reviewer Comment: More information could have been included regarding sample selection and demographic characteristics of the participants (e.g., socioeconomic status). Exclusion criteria were not provided.
Response: We thank the reviewer for this helpful feedback. We have revised the Methods section to include exclusion criteria and additional demographic context (lines 147–155): “Exclusion criteria included students with poor school attendance or academic performance, as the school requested that students who needed classroom time not be pulled from instruction.” “A total of 14 students participated in the co-design workshops. The sample included seven girls: White (n = 4, 57.1%), Hispanic (n = 2, 28.6%), and Black (n = 1, 14.3%). The seven boys included: White (n = 5, 71.4%) and Hispanic (n = 2, 28.6%).”
Reviewer Comment: It is unclear how participant feedback was analyzed.
Response: We appreciate this comment and have clarified the analytic process (lines 198–204). Participant feedback surveys (both open- and closed-ended items) were coded and summarized alongside workshop transcripts. Open-ended responses were incorporated into the thematic analysis, while closed-ended responses were summarized descriptively to provide additional context on feasibility, acceptability, and satisfaction. The revised text now states: “To complement these qualitative sources, participant feedback surveys administered after each session were analyzed descriptively and thematically. Open-ended responses were coded alongside workshop transcripts to capture adolescents’ direct reflections on feasibility, acceptability, and perceived relevance of proposed strategies. Closed-ended survey items were summarized quantitatively (e.g., frequencies, proportions) to provide contextual indicators of satisfaction and engagement. These multiple data streams were integrated during analysis to triangulate findings and strengthen interpretation.”
Reviewer Comment: What is the feedback survey and what did it ask? Where was the feedback used?
Response: Please see above comment.
Reviewer Comment: There are six themes; numerical data can be added to the findings.
Response: Thank you for the suggestion. Because this study used a qualitative co-design approach and reflexive thematic analysis, our goal was to prioritize depth, nuance, and context over numerical prevalence. Adding counts can imply population-level estimates or comparative weighting that are not appropriate for this analytic paradigm.
Reviewer Comment: Tables and figures can be added.
Response: We appreciate this suggestion. To enhance clarity and accessibility of the findings, we have added a figure to outline the themes identified and their overlap within the conceptual framework.
Reviewer Comment: The discussion becomes repetitive.
Response: We appreciate the reviewer’s perspective and carefully re-examined the Discussion section with this feedback in mind. While we recognize some thematic overlap across subsections, we intentionally structured the Discussion to align each set of findings with the guiding theoretical frameworks (BPNT, GCT, SCT, and SOCO). This organization necessarily revisits core constructs—autonomy, competence, relatedness, motivation, and social comparison—in different contexts (physical activity vs. social media), which may appear repetitive but ensures clarity and integration for readers. After review, we believe the current structure best supports transparency and theoretical grounding, so we retained it while ensuring transitions highlight distinct contributions within each subsection.
(see attached Word doc for table of author responses)

Reviewer 2 Report
Comments and Suggestions for Authors
The manuscript addresses an important topic, focusing on the intersection of rural adolescents’ physical activity, social media use, and mental health. The participatory co-design approach is innovative and ensures that youth voices are represented in the development of the Hoosier Sport Re-Social intervention. The paper is well written, theoretically grounded, and provides practical implications for intervention design.
In my opinion several aspects could be improved to further strengthen the manuscript:
Abstract and Introduction
- The introduction situates the study well in terms of rural health disparities, but the review of literature is strongly North American. Consider including more international studies on adolescent physical activity, social media, and mental health for global relevance.
Conceptual Framework (2.1)
- The integration of multiple theoretical frameworks (SDT, BPNT, SCT, SOCO) is a strength. However, the figure and caption (Figure 1) could be expanded to more clearly explain how these frameworks were operationalized in the study.
Methods
- The sample size (n = 14) is very small, which limits generalizability. While thematic saturation is claimed, please acknowledge this limitation more explicitly and discuss how findings could be transferred to other rural contexts.
- Reflexivity is not addressed. Please add a short discussion of how the researchers’ backgrounds or roles may have influenced data collection, coding, and analysis.
- The study relies exclusively on adolescents’ voices. Please justify the decision not to include parental or teacher perspectives, and reflect on how this might shape the findings.
- The coding process is well described, but it would be useful to specify how disagreements between coders were resolved beyond “collaborative discussion.”
Results
- In the “Social Comparison” theme, the link to gender differences is important. Consider elaborating briefly on whether boys also reported body image concerns, even if less frequently, to balance interpretation.
Limitations (4.7)
- Limitations are acknowledged, but reflexivity and lack of triangulation should also be listed explicitly.
Conclusions
- The conclusion emphasizes implications for intervention, but should also highlight more clearly the methodological contribution: the value of co-design with rural adolescents.
Overall Assessment
The study is promising and makes a valuable contribution to school-based public health research. With revisions to clarify methodological rigor, reflexivity, and transferability, this manuscript will be a strong candidate for publication.
Author Response
Reviewer Response – ijerph-3843797 (Reviewer 3)
|
Reviewer Comment |
Line Number(s) |
Response |
|
The introduction situates the study well in terms of rural health disparities, but the review of literature is strongly North American. Consider including more international studies on adolescent physical activity, social media, and mental health for global relevance. |
67-68 78 86-87 |
We thank the reviewer for this thoughtful suggestion. To enhance global relevance, we have expanded the Introduction to include recent international studies. |
|
The integration of multiple theoretical frameworks (SDT, BPNT, SCT, SOCO) is a strength. However, the figure and caption (Figure 1) could be expanded to more clearly explain how these frameworks were operationalized in the study. |
N/A |
Thank you for this helpful feedback. We have added an additional figure (Figure 3) to address this (located in results), as well as a figure caption: “Caption: The figure depicts the hypothesized pathway from guiding levels of influence (individual, interpersonal, community; SCT) to intervention strategies (sport participation, skill development with goal setting/monitoring, structured social support, education in physical and social-media literacy, and take-home activities) and then to theoretical mediating processes (satisfaction of competence, autonomy, and relatedness; BPNT within SDT). SCT constructs (self-efficacy, role modeling, social support) inform delivery at each level; GCT orients content toward intrinsic goals (well-being, mastery, enjoyment) rather than extrinsic (appearance/validation); and SOCO is addressed via media-literacy and norms-reflection components designed to reduce harmful upward comparison. Solid arrows indicate proposed causal links (levels → strategies → needs → behavior), and dotted arrows indicate mediating processes (need satisfaction) and cross-cutting influences (social comparison). These same constructs defined the deductive codebook and framework analysis matrix used in qualitative analysis.”
|
|
The sample size (n = 14) is very small, which limits generalizability. While thematic saturation is claimed, please acknowledge this limitation more explicitly and discuss how findings could be transferred to other rural contexts. |
528-534 |
We agree that the small sample size (n = 14) limits generalizability and have emphasized this more explicitly in the Limitations. Because this was qualitative co-design, our aim was depth and contextual insight rather than population inference. We now describe conditions under which transfer is plausible—e.g., rural schools with comparable resource constraints, PA opportunities, and social-media environments. “Accordingly, we frame our claims in terms of transferability rather than generalizability, providing thick description of the school, community context, and workshop procedures to enable readers to assess fit to other settings. These findings may be transferable to rural middle schools with similar structural features—e.g., limited access to affordable sport facilities, transportation barriers, and comparable social-media use patterns—particularly where school-based PA and health classes are the primary venues for programming.” |
|
Reflexivity is not addressed. Please add a short discussion of how the researchers’ backgrounds or roles may have influenced data collection, coding, and analysis. |
249-258 |
Thank you for this important suggestion. We have added a reflexivity statement in the Methods section to acknowledge how the research team’s backgrounds and roles may have influenced the co-design process and data interpretation. “The research team acknowledges that our professional and personal backgrounds influenced the design, facilitation, and interpretation of the co-design process. The two primary facilitators (JW and JG) had previously worked in the rural region on multiple school- and community-based projects and were familiar with the community context, which supported rapport and trust with participants. This familiarity may also have shaped assumptions during data collection and analysis. Other team members (AS, MK) contributed expertise in youth physical activity, mental health, and qualitative methodology, which informed both coding and interpretation. To reduce potential bias, we employed independent coding, peer debriefing, and iterative consensus meetings to critically reflect on how our positionalities influenced the analytic process.” |
|
The study relies exclusively on adolescents’ voices. Please justify the decision not to include parental or teacher perspectives, and reflect on how this might shape the findings. |
189-193 |
Thank you for this thoughtful comment. We intentionally centered adolescents’ voices in the co-design process, as their lived experiences were the primary focus for informing intervention development. To ensure adult perspectives were also considered, we convened an expert advisory committee composed of a teacher, two parents, a school board member, and the school principal. This group reviewed session materials, monitored feasibility, and provided feedback throughout implementation. We have clarified this rationale in the Methods. These findings represent one component of a larger Intervention Mapping study, in which the expert committee’s role is described in greater detail (manuscript submitted for publication). “In addition to adolescent participants, an expert advisory committee—comprising a teacher, two parents, a school board member, and the school principal—was convened to provide adult perspectives. The committee did not contribute data for analysis but reviewed session materials, monitored feasibility, and offered feedback throughout implementation.” |
|
The coding process is well described, but it would be useful to specify how disagreements between coders were resolved beyond “collaborative discussion.” |
237-239 |
Thank you for this feedback. We have added: “When coding disagreements occurred, they were first documented, then discussed in structured resolution meetings, and if consensus could not be reached, a third researcher adjudicated to finalize the code.” |
|
In the “Social Comparison” theme, the link to gender differences is important. Consider elaborating briefly on whether boys also reported body image concerns, even if less frequently, to balance interpretation. |
Results (Theme 6) |
Thank you for this thoughtful suggestion. We have added in the results: “While these concerns were more frequently and intensely described by female participants, some boys also reported pressures related to appearance and social standing, though often framed in terms of athletic performance, body size, or comparisons with peers on social media.” |
|
Limitations are acknowledged, but reflexivity and lack of triangulation should also be listed explicitly. |
560-566 |
We have adjusted the limitation to be more thorough: “We also acknowledge that the study was shaped by the research team’s positionality and backgrounds, which may have influenced facilitation and analysis, despite steps taken to reduce bias through independent coding and consensus-building. In addition, because the study relied primarily on adolescent voices and session transcripts, without triangulation across multiple data sources or stakeholder groups, findings should be interpreted with caution.” |
|
The conclusion emphasizes implications for intervention, but should also highlight more clearly the methodological contribution: the value of co-design with rural adolescents. |
589-592 |
Thank you for this feedback. We have added: “Methodologically, this work demonstrates the feasibility and added value of youth-centered co-design in rural school settings, offering a transferable model for participatory intervention development where resources are limited and local context is paramount.” |
Round 2
Reviewer 1 Report
Comments and Suggestions for Authors
Thank you for your revisions.